# miR-301a Deficiency Attenuates the Macrophage Migration and Phagocytosis through YY1/CXCR4 Pathway

**DOI:** 10.3390/cells11243952

**Published:** 2022-12-07

**Authors:** Jiawei Xu, Lanya Fu, Junyao Deng, Jiaqi Zhang, Ying Zou, Liqiang Liao, Xinrui Ma, Zhenlin Li, Yizhou Xu, Yuantao Xu, Shuyi Xu, Jingmin Liu, Xianghai Wang, Xiaodong Ma, Jiasong Guo

**Affiliations:** 1Department of Histology and Embryology, Guangdong Provincial Key Laboratory of Construction and Detection in Tissue Engineering, National Demonstration Center for Experimental Education, School of Basic Medical Sciences, Southern Medical University, Guangzhou 510515, China; 2Department of Neurology, Nanfang Hospital, Southern Medical University, Guangzhou 510515, China; 3Department of Spine Orthopedics, Zhujiang Hospital, Southern Medical University, Guangzhou 510280, China; 4Key Laboratory of Mental Health of the Ministry of Education, Guangdong-Hong Kong-Macao Greater Bay Area Center for Brain Science and Brain-Inspired Intelligence, Guangzhou 510515, China; 5Guangdong Province Key Laboratory of Psychiatric Disorders, Guangzhou 510515, China; 6Key Laboratory of Brain, Cognition and Education Sciences, Ministry of Education, Guangdong Key Laboratory of Mental Health and Cognitive Science, Center for Studies of Psychological Application, Institute for Brain Research and Rehabilitation, South China Normal University, Guangzhou 510631, China

**Keywords:** miR-301a, macrophage, migration, phagocytosis, YY1, CXCR4

## Abstract

(1) Background: the miR-301a is well known involving the proliferation and migration of tumor cells. However, the role of miR-301a in the migration and phagocytosis of macrophages is still unclear. (2) Methods: sciatic nerve injury, liver injury models, as well as primary macrophage cultures were prepared from the miR-301a knockout (KO) and wild type (WT) mice to assess the macrophage’s migration and phagocytosis capabilities. Targetscan database analysis, Western blotting, siRNA transfection, and CXCR4 inhibition or activation were performed to reveal miR301a’s potential mechanism. (3) Results: the macrophage’s migration and phagocytosis were significantly attenuated by the miR-301a KO both in vivo and in vitro. MiR-301a can target Yin-Yang 1 (YY1), and miR-301a KO resulted in YY1 up-regulation and CXCR4 (YY1′s down-stream molecule) down-regulation. siYY1 increased the expression of CXCR4 and enhanced migration and phagocytosis in KO macrophages. Meanwhile, a CXCR4 inhibitor or agonist could attenuate or accelerate, respectively, the macrophage migration and phagocytosis. (4) Conclusions: current findings indicated that miR-301a plays important roles in a macrophage’s capabilities of migration and phagocytosis through the YY1/CXCR4 pathway. Hence, miR-301a might be a promising therapeutic candidate for inflammatory diseases by adjusting macrophage bio-functions.

## 1. Introduction

Within most tissues, macrophages play a vital role in innate immunity, maintaining homeostasis, and producing immune responses to antigenic stimuli [1,2,3]. Once the tissue is injured, macrophages are recruited and migrate into the injury site to phagocytize the debris of degraded cells and invaded microbes so as to provide a favorable microenvironment for the following tissue repair [4,5,6]. Therefore, the capabilities of migration and phagocytosis are prerequisite for macrophages to play their role in tissue injury and repair. Previous studies indicate that many molecules including miRNAs are involved in regulating macrophage migration and phagocytosis. For example, miR-17 and miR-221 can promote macrophage migration via up-regulating the expression of integrin β1 [7,8], while miR-146a inhibits macrophage migration by suppressing IGF2BP1 and HuR [9].

The microRNA-301a (miR-301a) is well known as it has important roles in tumor biology. Much literature has reported that miR-301a is overexpressed in a variety of tumors and significantly promotes the proliferation and migration of the tumor cells [10,11,12]. Our recent studies figured out that genetic ablation of miR-301a could reduce pancreatic fibrosis and bleomycin-induced lung fibrosis [13,14]. Until now, scientists had some evidence about the character of miR-301a in macrophages. The existing literature suggested that several kinds of cancer-cell-derived exosomes might act on macrophage polarization attributing to its contained miR-301a [15], and Hsu et al. reported that miR-301a inhibition enhances the immunomodulatory functions of adipose-derived mesenchymal stem cells by induction of macrophage M2 polarization [16]. Huang et al. demonstrated that miR-301a regulates inflammatory cytokine expression in macrophages [17]. However, the potential role of miR-301a in the migration and phagocytosis capabilities of macrophages is still unknown.

In the present study, with miR-301a knockout (KO) mice, we firstly illustrated that miR-301a deficiency inhibits macrophage migration and phagocytosis in both contexts of sciatic nerve injury and liver injury models, as well as in the in vitro cultured macrophages. Regarding the mechanism, current data indicates that miR-301a plays a role in macrophage migration and phagocytosis through the YY1/CXCR4 signaling pathway, which is well known for its biofunction in cytoskeleton remodeling [18,19,20,21,22,23]. Overall, our findings raise the possibility that miR-301a might serve as a potential therapeutic target for inflammatory diseases by regulating macrophage migration and phagocytosis. This study also provided evidence to enrich our understanding of miR-301a’s role in macrophages’ bio-function.

## 2. Materials and Methods

### 2.1. Animals

All mouse lines were maintained on a C57BL/6J background and housed under standard conditions (22 ± 1 °C) in a specific pathogen-free animal facility with a 12/12 h light–dark cycle with water and food ad libitum at Southern Medical University. The Institutional Animal Care and Use Committee of Southern Medical University, P.R. China, approved all animal experiments. All efforts were made to minimize the number of animals and their suffering.

Generation of miR-301a^−/−^ (KO) mice in the C57BL/6J background has been described in our previous reports [13,24]. The homozygous KO mice have been bred for dozens of generations with stable inheritance in our laboratory. From the genotyping results, we can distinguish the wild type (WT) mice and the KO mice, as the DNA of the WT group showed at 370 bp while the KO group showed at 160 bp.

### 2.2. Preparation of Sciatic Nerve Injury and Liver Injury Models in Mice

Adult mice (8 weeks) were anesthetized with an intraperitoneal injection of 12 mg/mL tribromoethanol (180 mg/kg body weight). For nerve injury, the right sciatic nerve was bluntly exposed. A sciatic nerve transection injury model was established through a transection at 0.3 cm distal to the sciatic notch [25,26]. For the sham operation, the left sciatic nerve was only exposed without transection. For liver injury, the right lobe of liver was exposed and then penetrated vertically through a 1 mL syringe with a 26-gauge needle. Similarly, the liver of the sham group mice was only exposed liver tissue without damage.

### 2.3. Hepatic Macrophage Phagocytosis Assay with Trypan Blue Dye Injection In Vivo

Adult mice (8 weeks) with an intraperitoneal injection of 15 mg/mL 0.4% trypan blue dye (180 mg/kg body weight) after 24 h were anesthetized with an intraperitoneal injection of 12 mg/mL tribromoethanol (180 mg/kg body weight). The liver was harvested from mice after transcardial perfusion with 4% paraformaldehyde (PFA, Macklin, P804536) and post-fixed in 4% PFA for 24 h. The mid-belly of each liver was trimmed for routine paraffin embedding, and then transversally sectioned with a thickness of 7 μm for routine eosin staining to show the phagocytosis of trypan blue.

### 2.4. Culture of Bone-Marrow-Derived Macrophages and Pharmacological Treatment

Macrophages were isolated and cultured as described previously [27]. Briefly, bone marrow cells were flushed out with Dulbecco’s modified Eagle’s medium (DMEM)/F12 (Gibco, Grand Island, NY, USA, 11330057) from the tibia and femur of the wide type (WT) and miR-301a knockout (KO) mice after euthanasia. The bone marrow cells were cultured in DMEM/F12 containing 10% fetal bovine serum (FBS, BioChannel Biological Technology, BC-SE-FBS01C), antibiotics (100 U/mL penicillin, 100 mg/mL streptomycin, Gibco, Grand Island, NY, USA15140122), and 10 ng/mL macrophage colony stimulating factor (MCSF, R&D, 416-ML-010) at 37 °C and under 5% CO_2_ for 7 days.

To investigate the roles of CXCR4 in macrophage migration and phagocytosis, 10 μM WZ811 (CXCR4 inhibitor) and 1 μM ATI2341 (CXCR4 agonist) were added into the cultured medium in WT and KO macrophages and maintained for 48 h.

### 2.5. Culture of RAW264.7 Cells

The mouse monocyte-macrophage line RAW264.7, purchased from the American Type Culture Collection (ATCC, Rockville, MD, USA), was maintained in (DMEM)/F12 supplemented with 10% FBS containing 10% FBS, 100 U/mL penicillin, and 100 mg/mL streptomycin at 37 °C and under 5% CO_2_ [27].

### 2.6. YY1 siRNAs Transfections

RAW264.7 cells or primary cultured macrophages were transfected with 40 nM YY1 siRNAs or negative control using LipoRNAi transfection reagent (Beyotime, Shanghai, China, C0535), respectively. The sequences of siRNAs are shown in Table 1. The serum-free medium, siRNA, and LipoRNAi transfection reagent were mixed directly, incubated at room temperature for 20 min, and then were directly added to the cell culture vessel. The transfected cells were incubated at 37 °C and 5% CO_2_ for 48 h. Subsequently, a Western blot was performed to determine the expression of YY1 as well as other aim proteins in cells treated with siRNA. Additionally, these cells were used in follow-up experiments.

### 2.7. Myelin Debris Preparation for the In Vitro Phagocytosis Assay

Myelin debris were produced using mouse brains and spinal cords as described elsewhere [25]. In brief, brains and spinal cords were isolated from wild type mice aged 8–10 weeks after euthanasia by cervical dislocation and then shattered into tiny particles by sonication. Tissue debris was rinsed with Milli-Q water three times by centrifugation for 15 min at 4 °C at 14,462 g. Finally, myelin debris were resuspended with Hank’s balanced salt solution (HBSS, Gibco, Grand Island, NY, USA, C14175500BT) at a concentration of 100 mg/mL and stored at −80 °C.

### 2.8. Phagocytic Capability Assay in the Cultured Macrophages

Phagocytic capability of the cultured macrophages was assessed by the ingestion of lumispheres or myelin debris [28]. Briefly, macrophages were seeded on the coverslips in 24-well plates at a density of 1 × 10^5^ cells/well. The next day, 0.1 mg/mL fluorescent lumispheres (1 µm diameter, BaseLine Chromtech, Tianjin, China, 7-3-0100) or 1 mg/mL myelin debris were added into the culture for 3 h or 24 h, respectively. After being rinsed three times with HBSS to remove the attached lumispheres or myelin debris from the cell surface, the cells were fixed with 4% PFA and immunostained with F4/80 antibody (1:300, Abcam, Cambridge, UK, ab6640) to identify the outlines of the macrophages. The number of lumispheres ingested by each cell was counted under a fluorescent microscope (Leica, Wetzlar, Germany). To observe the ingestion of myelin debris, F4/80 was co-stained with oil-red-O (ORO, Solarbio, Beijing, China, G1262). Myelin basic protein (MBP, 1:100, Biolegend, Beijing, China, SMI-99) was elevated by Western blotting.

### 2.9. Migration Assay in the Cultured Macrophages

The migration of the cultured macrophages was assessed by a transwell assay using 6.5 mm transwell chambers (8 µm pores, Corning Costar, New York, NY, USA, 3422) as described previously [25,29]. After the chambers were pretreated with 0.1 mg/mL Poly-L-lysine hydrobromide (PLL, Sigma-Aldrich, St. Louis, MO, USA, P1274) solution or culture medium, 1 × 10^5^ macrophages in 100 µL of DMEM/F12 containing 1% FBS were seeded into the upper chamber, and the lower chamber was filled with 600 µL DMEM/F12 containing 10% FBS and 1 mg/mL myelin debris without cells. The macrophages were allowed to migrate for 18 h, and then the chambers were fixed with 4% PFA for 20 min. After careful removal of the cells on the upper surface with a cotton swab, the cells adhered to the lower surface of the transwell membrane were stained with 0.1% crystal violet (Leagene, Beijing, China, DZ0055) for 30 min. Then, five images of each membrane (the center and four quadrants) were captured under an inverted microscope (Leica, Wetzlar, Germany) for quantification.

### 2.10. Immunohistochemistry

The collected sciatic nerves and livers were subjected to perform the immunohistochemistry as previous reported [28,30]. Briefly, after dehydration in 30% sucrose overnight, the tissues were embedded in optimum cutting temperature compound (OCT, CellPath, UK1861102) for frozen sectioning. The sections were permeabilized with 0.5% Triton X-100 (Sigma, St. Louis, MO, USA, X100-100 ML) for 30 min, blocked with 5% fish gelatin (Sigma, St. Louis, MO, USA, G7041-500 G) containing 0.3% Triton X-100 at room temperature (RT) for 1 h, then incubated with the F4/80 antibody (1:300) at 4 °C overnight. Alexa 488 and/or 568 fluorescent-conjugated secondary antibodies were applied at RT for 2 h. Finally, 1 µg/mL 4′,6-diamidino-2-phenylindole (DAPI, Sigma, St. Louis, MO, USA, D8417-10 MG) was incubated for 2 min to counterstain cell nuclei. To evaluate the recruited macrophages around the injury site, the F4/80 fluorescence intensity was measured in a rectangle image with the size of 2 mm × 4 mm in the nerve injury site, and 1 mm × 0.5 mm in the liver injury site.

### 2.11. Oil Red O (ORO) Staining

For assessing the phagocytosis of macrophages, ORO staining was performed [31]. Briefly, the ORO staining solution was prepared as a mixture of 0.3% ORO (dissolved in 60% 2-propanol, Solabio, Beijing, China, G1262) and deionized water (ratio = 3:2). After being rinsed with 0.01 M PBS and 60% isopropanol, the sections were incubated in the ORO solution at 37 °C for 15 min and then rinsed in 60% isopropanol and 0.01 M PBS before routine mounting.

### 2.12. Western Blotting

For Western blotting, subjected cells were washed twice with PBS, scraped, and lysed in a RIPA buffer (Fdbio science, Hangzhou, China, FD009) containing a protease inhibitor cocktail (1:100, Fdbio, Hangzhou, China, FD1001). Lysates were incubated on ice for 30 min and centrifuged (14,462 g, 20 min, 4 °C) to collect the supernatant. Extracts were separated in an SDS-PAGE sample buffer (Fdbio science, Hangzhou, China, FD006) on 10% SDS-PAGE gels and transferred to a polyvinylidene difluoride membrane (PVDF, Millipore, Burlington, MA, USA, IPVH0010). Blots were blocked with 5% BSA for 1 h and incubated with primary antibodies overnight at 4 °C and then incubated with HRP-conjugated secondary antibodies at RT for 2 h. The following primary antibodies were used: rabbit anti-GAPDH (1:2000, Abcam, Cambridge, UK, ab8245), mouse anti-MBP (1:500, Biolegend, Beijing, China, SMI-99), rabbit anti-YY1 (1:1000; Abcam, Cambridge, UK, ab109237), rabbit anti-CXCR4 (1:500, Abcam, Cambridge, UK, ab181020). The secondary antibodies were as follows: goat anti-rabbit IgG (1:2000, Invitrogen, Pittsburgh, PA, USA, 31460), or goat anti-mouse IgG (1:2000, Invitrogen, Pittsburgh, PA, 32430).

The Western blotting for each protein was repeated three times for each test. Immunoreactive protein bands were detected and imaged by enhanced chemiluminescence (ECL, Millipore, Burlington, MA, USA) using a Lumazone system (Roper, Trenton, NJ, USA). The integrated optical density (IOD) of each lane was quantified with the Image J software 1.50i (Media Cybernetics, Silver Spring, MD, USA), and the expression levels of the targeted proteins were calculated by dividing the IOD of the targeted proteins by the IOD of the GAPDH.

### 2.13. Statistical Analysis

All values are presented as the mean ± standard error of the mean (SEM), and the SPSS 23.0 software (IBM, Armonk, NY, USA) was used for the statistical analysis. Differences between the two groups were analyzed using the Student’s *t*-test. A one-way ANOVA (Bonferroni post hoc comparison) was used to determine the statistical significance of the differences among multiple groups, and the *p*-values are indicated by * *p* < 0.05, ** *p* < 0.01, *** *p* < 0.001. A *p*-value of <0.05 was considered statistically significant.

## 3. Results

### 3.1. miR-301a Deficiency Attenuates the Migration of Macrophages

Under immunostaining, a few of the F4/80-positive macrophages were found scattering in the sciatic nerves and liver tissues, and the number of macrophages in the nerve and livers had no significant differences between wild type (WT) and KO mice. To explore the potential role of miR-301a in macrophages migration, the sciatic nerve transection injury model (Figure 1A,B) and liver needle stick injury model (Figure 1A) were established in miR-301a KO and WT mice and then the number of macrophages recruited in the lesion area was quantified (Figure 1B). As per our previous reports [27,31], a large number of macrophages were observed to gather in the sciatic nerve distal segment at 5 days post-injury (dpi) in the nerve transection injury model. Interestingly, the density of macrophages in the injured sciatic nerves of the KO group was significantly lower than that of the WT group (Figure 1B). Meanwhile, in the livers with the needle stick injury, they also clearly showed that macrophages aggregate around the injury site, and the KO mice showed fewer macrophages in the injured liver (Figure 1C). These results indicated that macrophage recruitment in the injured tissue was attenuated in the miR-301a-deficient mice.

Since macrophage recruitment depends on both macrophages’ migration capability and chemokine concentration in the lesion area, in order to exclude the potential influences of miR-301a on the chemokine production by other cells, isolated macrophages in cultures were utilized to perform in vitro transwell assays to detect the macrophage migration capability again (Figure 1D). As shown in Figure 1D, the number of macrophages that migrated through the transwell chamber was significantly decreased in the miR-301a KO group.

### 3.2. miR-301a Deficiency Attenuates the Phagocytosis of Macrophages

To explore whether miR-301a deficiency in macrophages might affect their phagocytosis capability, we detected myelin debris engulfment in the injured nerve and the ingestion of trypan blue dye in the hepatic macrophages. Based on the F4/80 immunofluorescence and oil-red-O (ORO) double staining images (Figure 2A), quantification showed that the percentage of macrophages engulfed with ORO-positive myelin debris (Figure 2B) as well as the area ratio of ORO-positive myelin debris droplets in each macrophage (Figure 2C) in miR-301a KO nerves were significantly lower than those in WT nerves. Meanwhile, 24 h after the intraperitoneal injection of trypan blue dye, the dye-ingested macrophages were easy to find in the liver (Figure 2D), by which, the number of dye-ingested macrophages (Figure 2D) and the total area of trypan blue (Figure 2F) in the miR-301a KO group were significantly decreased than those of the WT group.

The above findings from the in vivo experiments indicated that miR-301a deficiency attenuates macrophages’ phagocytosis. To verify this view, the in vitro experiments of co-culturing macrophages with fluorescent lumispheres (Figure 2G,H) or myelin debris (Figure 2I–M) were implemented. Quantification of the immunostained cultures showed that the numbers of lumispheres (Figure 2H) or the ORO-positive myelin debris engulfed in each macrophage (Figure 2J,K) of the KO group were significantly less than that of the WT group. Moreover, myelin debris phagocytized in the macrophages were also detected by Western blotting with MBP antibody, since naïve macrophage do not express MBP which is one of specific protein of myelin. The blots and quantification illustrated that there was less MBP protein in the KO group (Figure 2L,M).

### 3.3. miR-301a Regulates the Expression of YY1 and CXCR4 in Macrophages

To explore the potential mechanism of miR-301a in macrophage migration and phagocytosis, we firstly conducted a Targetscan database analysis. Herein, miR-301a was found to directly target YY1 (Figure 3A), which can negatively regulate CXCR4 expression. Therefore, Western blotting was done and revealed that the expression levels of YY1 were increased while CXCR4 levels were decreased in the KO group compared to those of the WT group (Figure 3B–D).

Furthermore, YY1 siRNA interference was implemented to reveal the relationship between YY1 and CXCR4 in macrophages. Firstly, three sets of YY1 siRNAs (siYY1-1, siYY1-2, and siYY1-3) were developed and the knockdown efficiency was detected with RAW264.7 cells (a macrophage cell line, Figure 3E,F). Then, the best one (siYY1-3) was chosen for the subsequent experiments. After the YY1 protein level was knocked down in the primary cultured macrophages, the expression levels of CXCR4 in both the miR-301a KO and WT groups were significantly increased (Figure 3G–I).

### 3.4. miR-301a Regulates the Migration of Macrophages through the YY1/CXCR4 Pathway

To make sure whether miR-301a regulates the migration of macrophages through the YY1/CXCR4 pathway, siRNA targeting YY1 gene, the CXCR4 specific inhibitor (WZ811) [32,33], and a specific agonist (ATI2341) [34,35] were individually used to assess their effects on the migration capability in both miR-301a KO and WT macrophages with the transwell assay. Collected results with statistical analysis illustrated that the number of migrated macrophages through the transwell was significantly increased after siYY1 transfection in both WT and miR-301a groups (Figure 4A,B). Moreover, CXCR4-specific inhibition with WZ811 exacerbated the macrophages’ lower migration capability while CXCR4 activation with ATI2341 can significantly reverse the down-regulation effect of miR-301a on the macrophages’ migration (Figure 4C,D).

### 3.5. miR-301a Regulates the Phagocytosis of Macrophages through the YY1/CXCR4 Pathway

After siYY1 transfection, the capability of macrophage phagocytosis was also assessed. The overall data showed that siYY1 transfection resulted in more lumispheres which could be engulfed in each macrophage in both the WT and KO groups (Figure 5A,B). Meanwhile, the amount of myelin debris in the macrophages was also measured with MBP Western blotting and showed much higher levels in the siYY1-treated groups (Figure 5C,D). A higher ratio of ORO-positive macrophages as well as ORO-positive areas in each macrophage were detected in the siYY1-treated groups during the myelin debris ingestion testing (Figure 5E–G). Subsequently, WZ811 or ATI2341 was used to test the effects of CXCR4 inhibition or activation on the macrophages’ phagocytosis, and the results demonstrated that WZ811treatment attenuated the fluorescent lumispheres and myelin debris phagocytosis, while ATI2341 treatment reversed the phagocytosis weakness in the miR-301a KO macrophages (Figure 6).

## 4. Discussion

In past years, many studies have demonstrated that miR-301a is highly overexpressed in a variety of tumors. It regulates many biological functions of tumor cells, such as proliferation, migration, invasion, and colony formation, as well as the expression of pro-inflammatory factors [12,14,36,37]. Moreover, the exosomes derived from the tumor cells might deliver miR-301a to influence other cells including macrophages’ polarization [38]. Huang et al. [17] discovered that down-regulation of miR-301a suppresses inflammatory cytokine expression in macrophages, which also confirmed the regulatory role of miR-301a in macrophages in inflammatory responses. Except for the polarization and inflammatory cytokine expression, macrophages playing roles in tissue injury and regeneration also depend on their capabilities of migration into the lesion area, phagocytosis for clearing the degraded tissue, and invading foreign items [4]. Considering more and more reports emphasize the high expression level of miR-301a in tumor cells and suggest using miR-301a inhibitors to treat tumors, no matter whether the tumor cells expressed miR-301a, miR-301a inhibitors would inevitably affect the macrophages; therefore, we believe it is urgently necessary to reveal the exact roles of miR-301a in macrophages.

Recently, we reported that RhoA-conditional knockout in macrophages resulted in fewer macrophages migrating into the injured sciatic nerve and weaker capability of phagocytosis for removing the debris of degenerated axons and myelin during the Wallerian degeneration [25]. We believe the sciatic nerve injury can also be used as a credible animal model to test the effects of miR-301a KO on macrophage migration and phagocytosis. In order to make the conclusion more convincing, another model of liver injury in the miR-301a KO and WT mice was included. Based on these two models, we can confidently declare that the macrophages’ capabilities of migration and phagocytosis are significantly attenuated in the miR-301a KO mice in comparison to the WT mice.

After the phenomena in the in vivo experiments were revealed, the in vitro cultured macrophages were used to confirm the effects of miR-301a KO and then to explore the potential mechanisms. It is generally considered that both migration and phagocytosis are inseparable from the depolymerization, remodeling, and extension of the cytoskeleton. Our recent study indicated that RhoA plays role in macrophage migration and phagocytosis through the ROCK/MLCK pathway [25]. This pathway is well known for regulating cytoskeleton reorganization [39,40]. Based on the clues of cytoskeleton regulating pathways, we conducted a Targetscan database analysis and discovered that miR-301a can directly target the transcription factor YY1, which participates in various biological functions, including migration, invasion, embryogenesis, cell proliferation, and so on [41,42,43,44]. Previous studies had demonstrated that YY1 inhibits CXCR4 by binding to the upstream region of the CXCR4 promoter [45]. As an important chemokine receptor, CXCR4 is reported that it can regulate tumor cell migration, invasion, and cytoskeleton rearrangements through activating RhoA [19,20,21,22,23]. Additionally, a recent study demonstrated that CXCR4 interacts directly with the cytoskeleton and plays an important role in neuronal migration [18]. Moreover, the process of phagocytosis is also highly dependent on cytoskeleton rearrangements [46]. By combining these literatures and our previous finding, we speculated that miR-301a might play role in cytoskeleton rearrangements to regulate macrophage migration and phagocytosis through the YY1/CXCR4 pathway. Intriguingly, the present data proved that miR-301a deficiency results in up-regulation of YY1 and down-regulation of CXCR4, and siYY1 can reverse the down-regulation of CXCR4 in the miR-301a KO cells. In short, these data demonstrate there is a miR-301a/YY1/CXCR4 signaling pathway existing in macrophages.

Therefore, the next question is whether miR-301a KO affects macrophage migration and phagocytosis via the YY1/CXCR4 signaling pathway. To figure out this issue, YY1-siRNA, CXCR4-specific inhibitor (WZ811), and an agonist (ATI2341) were individually used to assess their effects on the migration and phagocytosis capabilities in both miR-301a KO and WT macrophages. The overall data demonstrated that the miR-301a-KO-derived lower capabilities of migration and phagocytosis were significantly rescued after siYY1 transfection or ATI2341 treatment, while WZ811 treatment resulted in more severe attenuation of macrophage migration and phagocytosis.

## 5. Conclusions

The collective data of the present study indicated that miR-301a KO may affect macrophages’ migration and phagocytosis via the YY1/CXCR4 pathway. Although further molecular mechanisms need to be explored, this study provided experimental evidence to enrich our understanding of miR-301a’s role in macrophages, and raise the possibility that miR-301a might serve as a potential therapeutic target for inflammatory diseases by regulating macrophage migration and phagocytosis.

## Figures and Tables

**Figure 1 cells-11-03952-f001:**
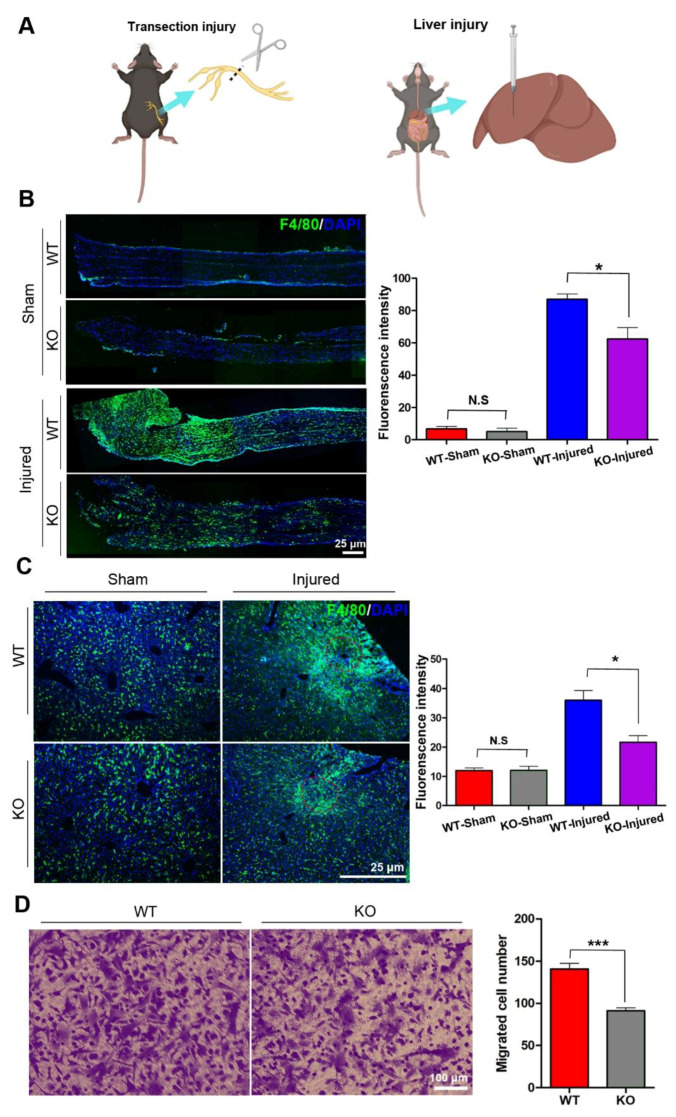
miR-301a knockout inhibits the migration of macrophages in vivo and in vitro. (**A**) Sciatic nerve transection injury model and liver needle stick injury model. (**B**) Immunohistochemistry and quantitative analysis in the longitudinal sections of transected sciatic nerve’s distal trunk at 5 dpi, which shows the recruited macrophages in the KO group are less than the WT group. Scale bar: 25 μm. (**C**) Immunohistochemistry and quantitative analysis in the needle stick injured liver showing that the migrated macrophages to the injury site in the KO group are less than the WT group. Scale bar: 25 μm. (**D**) Transwell assay and quantitative analysis showing the number of migrated macrophages in the KO group is significantly less than the WT group in vitro. Data are expressed as the mean ± SEM (*n* = 6 for each group, N.S: *p* ≥ 0.05, * *p* < 0.05, *** *p* < 0.001).

**Figure 2 cells-11-03952-f002:**
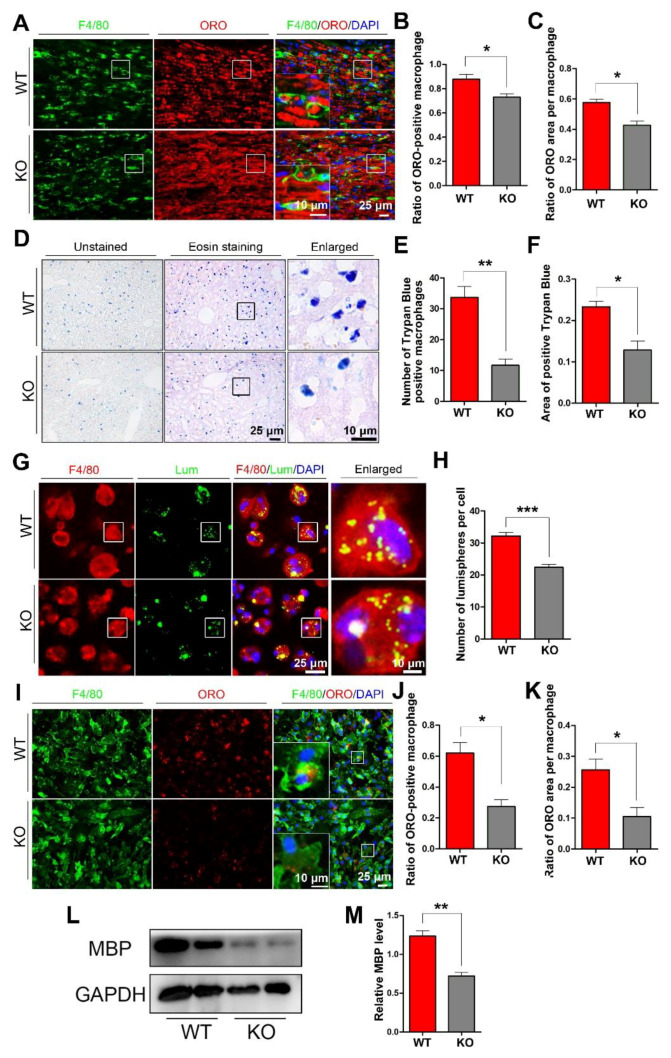
miR-301a knockout inhibits the phagocytosis of macrophages in vivo and in vitro. (**A**) F4/80 and ORO double staining in the transection-injured sciatic nerve at 5 dpi. (**B**,**C**) Quantitative analysis showing that the ratio of ORO+ macrophages and the ORO+ area in each macrophage are significantly decreased in the KO group compared to the WT group. (**D**–**F**) Eosin staining of liver tissue after trypan blue dye injection showing the phagocytosis of hepatic macrophages. (**G**) F4/80 immunocytochemistry with the fluorescent lumispheres and (**H**) quantification of the number of lumispheres per macrophage in vitro. (**I**–**K**) F4/80 and ORO double immunocytochemistry staining and quantitative analysis after the macrophages cultured with myelin debris in vitro. (**L**,**M**) Western blots and quantification showing that MBP expression in the KO group cultured with myelin debris is less than those in the Cre group in vitro. Data are expressed as the mean ± SEM (*n* = 6 for each group, N.S: *p* ≥ 0.05, * *p* < 0.05, ** *p* < 0.01, *** *p* < 0.001).

**Figure 3 cells-11-03952-f003:**
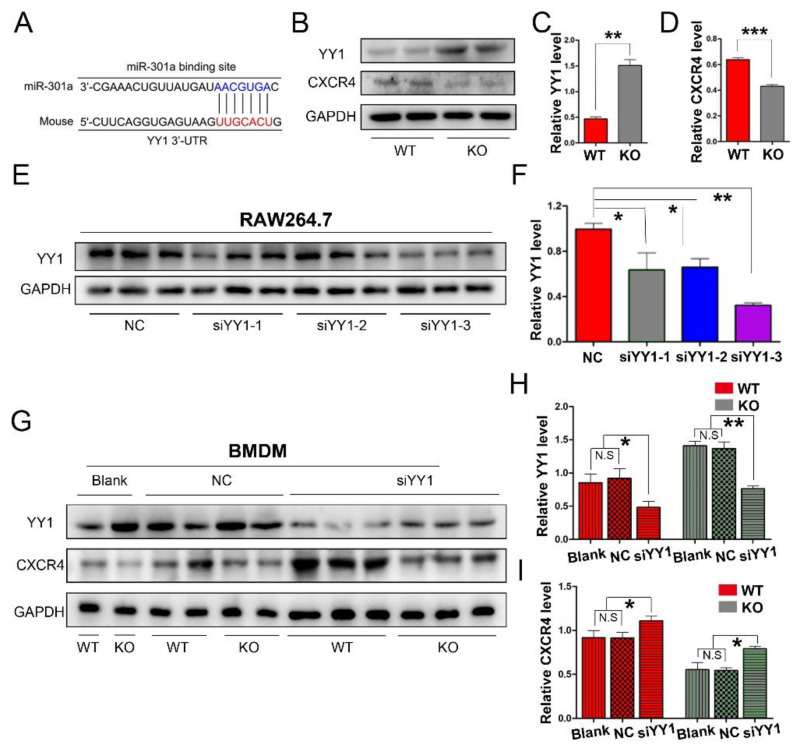
miR-301a KO in macrophages enhances the expression of YY1 and inhibits the expression of CXCR4. (**A**) The Targetscan database analysis suggested that miR-301a directly targets YY1. (**B**) Western blots and (**C**,**D**) statistical analysis illustrate that miR-301a knockout in macrophages up-regulates the protein level of YY1 and reduces the protein level of CXCR4 compared to WT group. (**E**) Western blots and (**F**) quantification analysis of YY1 expression inhibition by using YY1 siRNAs in RAW264.7 cells. (**G**) Western blots and (**H**,**I**) quantification of YY1 and CXCR4 expression in siYY1-3-transfected BMDM illustrate that the up-regulation of YY1 and the down-regulation of CXCR4 by miR-301a knockout are reversed by down-regulation of YY1. Data are expressed as the mean ± SEM (*n* = 3 for each group, N.S: *p* ≥ 0.05, * *p* < 0.05, ** *p* < 0.01, *** *p* < 0.001).

**Figure 4 cells-11-03952-f004:**
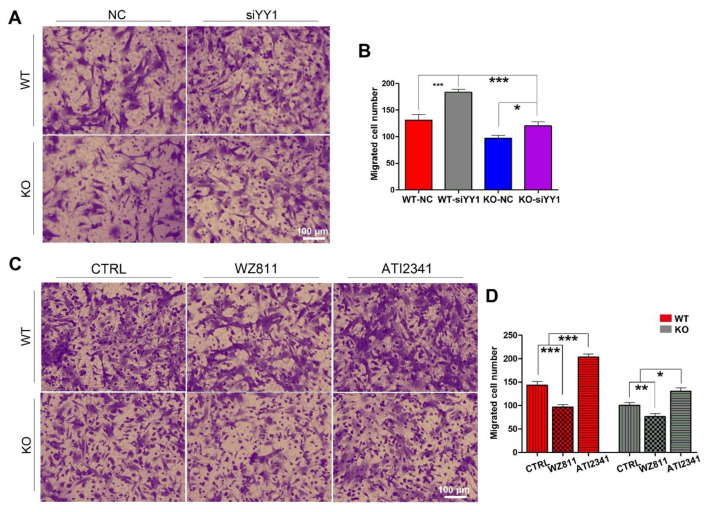
Transwell assay and quantifications show the effects of siYY1 transfection, WZ811, and ATI2341 treatment on the macrophage migration. (**A**) Transwell assay and (**B**) quantitative analysis showing the number of migrated macrophages from KO–siYY1 group is significantly more than the KO–NC group. (**C**) Transwell assay and (**D**) quantitative analysis showing the number of migrated macrophages from the KO–ATI2341 group can be reversed compared to KO–CTRL group. Data are expressed as the mean ± SEM (*n* = 3 for each group, N.S *p* ≥0.05, * *p* < 0.05, ** *p* < 0.01, *** *p* < 0.001).

**Figure 5 cells-11-03952-f005:**
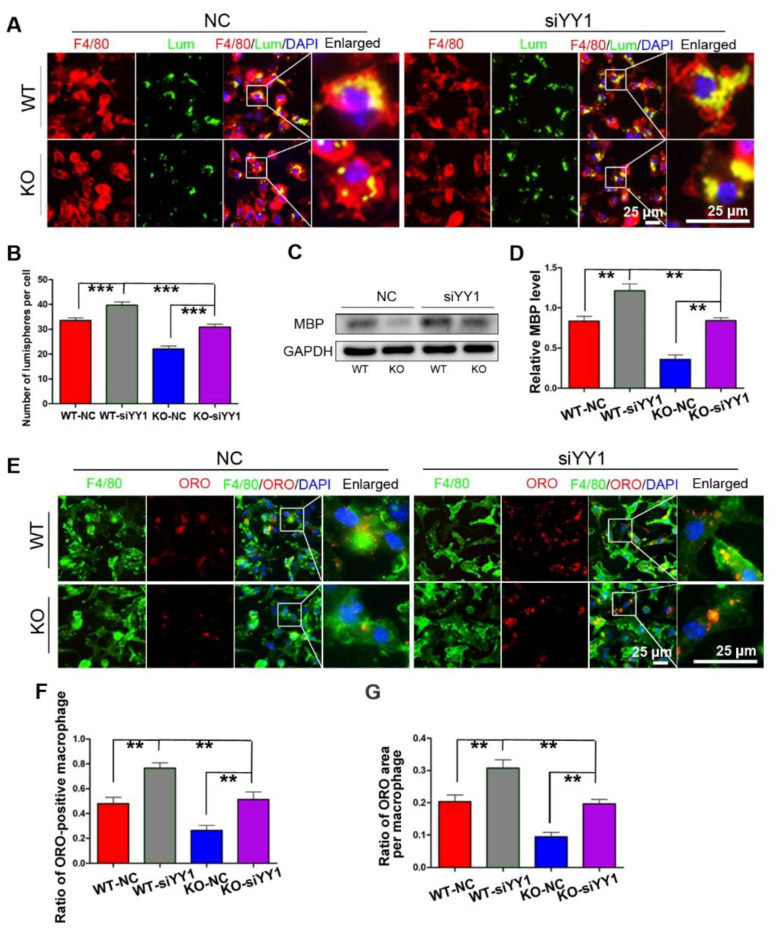
siYY1 transfection inhibits macrophage phagocytosis. (**A**) F4/80 immunocytochemistry with the fluorescent lumispheres and quantitative analysis (**B**). (**C**) Western blots and (**D**) quantification showing that MBP expression in the KO–siYY1 group cultured with myelin debris is more than those in the KO–NC group in vitro. (**E**) F4/80 and ORO double immunocytochemistry staining and (**F**,**G**) quantification of the ratio of ORO+ macrophages and the ORO+ area in each cell after the macrophages cultured with myelin debris in vitro. Data are expressed as the mean ± SEM (*n* = 3 for each group, N.S *p* ≥ 0.05, ** *p* < 0.01, *** *p* < 0.001).

**Figure 6 cells-11-03952-f006:**
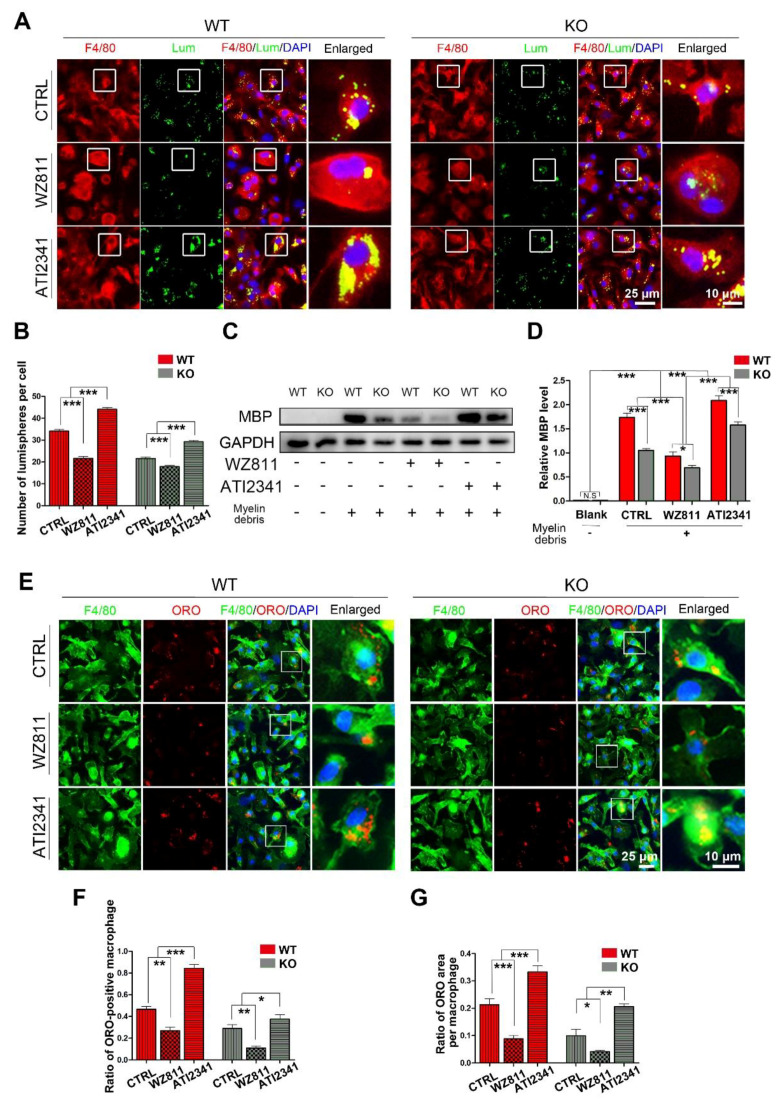
The CXCR4 inhibitor (WZ811) and agonist (ATI2341) regulate macrophage phagocytosis. (**A**) F4/80 immunocytochemistry with the fluorescent lumispheres and (**B**) quantitative analysis. (**C**,**D**) Western blots and quantification showing the ingested MBP expression in vitro. (**E**) F4/80 and ORO double immunocytochemistry staining and (**F**,**G**) quantification of the ratio of ORO+ macrophages and the ORO+ area in each cell after the macrophages cultured with myelin debris in vitro. Data are expressed as the mean ± SEM (*n* = 3 for each group, N.S *p* ≥0.05, * *p* < 0.05, ** *p* < 0.01, *** *p* < 0.001).

**Table 1 cells-11-03952-t001:** Sequences of siRNA.

siRNA	Sense (5′-3′)	Antisense (5′-3′)
siYY1-1	GCGACGACGACUACAUAGATT	UCUAUGUAGUCGUCGUCGCTT
siYY1-2	GAAGAUGAUGCUCCAAGAATT	UUCUUGGAGCAUCAUCUUCTT
siYY1-3	CGACGGUUGUAAUAAGAAGUU	AACUUCUUAUUACAACCGUCG
Negative control	UUCUCCGAACGUGUCACGUTT	ACGUGACACGUUCGGAGAATT

## Data Availability

All data supporting reported results can be found in the manuscript.

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
