# Peer review of "miR-301a Deficiency Attenuates the Macrophage Migration and Phagocytosis through YY1/CXCR4 Pathway"

_cells, 2022, doi:10.3390/cells11243952_

Round 1
Reviewer 1 Report
The authors characterized miR-301a as a potential therapeutic target for inflammatory diseases by regulating macrophage migration and phagocytosis.
Preliminary data underline the role of miR-301a in knockout (KO) and wild-type (WT) mice and also in mouse cells; evaluating the role of macrophage migration and phagocytosis through the YY1 / CXCR4 pathway. The article is written quite correctly although an appropriated editing of English language and style required and some points should be addressed:
1. Authors should specify the full names of some acronyms used to enable the reader to understand and interpret the content. For example, ORO is specified in the materials and methods but not in the introduction.
2. It would be better to add more findings about CXCR4 and its function. In general improve the introduction section.
3.Since the goal is to characterize miR-301a as a potential target for inflammatory diseases, The authors must support their suggestion although also anticipated from other papers (e.g.PMID: 23808420. They must correlate the role of miR-301a in inflammation via YY1 / CXCR4 pathway evaluating also immuno/inflammatory mediator as TNF-α, INF-γ, and IL-1β expression (e.g. by ELISA, western blot assay, RT-PCR e/o IF).
4. About the phagocitosis, the authors should evaluate also the intracellular Ca++ signalling or ATP production.
5. The authors could confirm some points also in human non-tumor macrophage cell lines such as hMDM (human monocyte-derived macrophages or also in human cancer macrophages.
Author Response
The authors characterized miR-301a as a potential therapeutic target for inflammatory diseases by regulating macrophage migration and phagocytosis.
Preliminary data underline the role of miR-301a in knockout (KO) and wild-type (WT) mice and also in mouse cells; evaluating the role of macrophage migration and phagocytosis through the YY1 / CXCR4 pathway. The article is written quite correctly although an appropriated editing of English language and style required and some points should be addressed:
Response: We sincerely appreciate your great comments. The manuscript has been revised thoroughly including the English language and style. The point-by-point responses for other comments are included as following.
- Authors should specify the full names of some acronyms used to enable the reader to understand and interpret the content. For example, ORO is specified in the materials and methods but not in the introduction.
Response: Thanks. These details have been added into the revised manuscript.
- It would be better to add more findings about CXCR4 and its function. In general improve the introduction section.
Response: Thanks for your suggestion. We briefly introduced CXCR4’s function in the introduction section and added more information about it in the discussion section.
3.Since the goal is to characterize miR-301a as a potential target for inflammatory diseases, The authors must support their suggestion although also anticipated from other papers (e.g.PMID: 23808420. They must correlate the role of miR-301a in inflammation via YY1 / CXCR4 pathway evaluating also immuno/inflammatory mediator as TNF-α, INF-γ, and IL-1β expression (e.g. by ELISA, western blot assay, RT-PCR e/o IF).
Response: Thanks a lot for your suggestion. We have revised the related description in the introduction and discussion to state this issue more clearly. Indeed, there are some studies, including you mentioned literature which was cited in the revised manuscript, indicate that miR-301a might play roles in the polarization of macrophages. Therefore, we did not detect the polarization in present study but focused on the migration and phagocytosis which are also critical in the process of macrophages participating in the inflammatory response. Due to the same reason, we did not detect the expression of immuno/inflammatory mediator as TNF-α, INF-γ, and IL-1β which are mainly regulated by the polarization of macrophages. We sincerely wish to get your understanding for this explain. Thank you very much!
- About the phagocitosis, the authors should evaluate also the intracellular Ca++ signalling or ATP production.
Response: We agree with that intracellular Ca2+ signaling and ATP production play roles in the phagocytosis, and there are references indicate that activation of CXCR4 can trigger the increase of intracellular calcium concentration as well as the intracellular ATP level. As well known, both migration and phagocytosis are highly dependent on the ATP level. However, whether the intracellular calcium is necessary for phagocytic ingestion has been debated much. Although it is now general accepted that a rise in intracellular calcium is an early event that accompanies phagocytosis, particle ingestion appears to be largely Ca2+-independent (P. Nunes, N. Demaurex: The role of calcium signaling in phagocytosis. J Leukoc Biol 2010, 88:57-68.). Therefore, we did not involve the intracellular Ca2+ signaling or ATP production in present study. Alternatively, we mainly discussed the potential mechanism of miR-301a in the migration and phagocytosis of macrophages through the YY1/CRCR4 to regulate cytoskeleton remodeling, which is another well-accepted bioeffect of CXCR4 signaling. - The authors could confirm some points also in human non-tumor macrophage cell lines such as hMDM (human monocyte-derived macrophages or also in human cancer macrophages.
Response: Thanks for this suggestion. However, it is very difficult for us to do the experiments on the human cells. Firstly, present study was mainly executed on miR-301a knockout mice. If we want to confirm the findings in the knockout mice, we have to develop the miR-301a knockout human cell lines. Secondly, since we did not include these cells in our application for the approvement of this study. If we want to supplement this part of the experiment, we must reapply for an ethical approval, this will be a time-consuming work. Considering that the absence of this part of data will not affect our conclusion, we sincerely apply for an exemption for this comment.
Reviewer 2 Report
This is an interesting article to understand role of miR-301a in macrophage migration and phagocytosis. They have shown that YY1/CXCR4 signaling pathway is involved. The study is well designed and results are clear.
Figure 4C: Authors show the same picture in WT macrophages in control condition and treated with WZ811!
Authors must add in legend the number of mice or field used to perform statistical analysis (n= ).
Author Response
This is an interesting article to understand role of miR-301a in macrophage migration and phagocytosis. They have shown that YY1/CXCR4 signaling pathway is involved. The study is well designed and results are clear.
Response: Thank you very much. The point-by-point responses for your comments are included as following.
Figure 4C: Authors show the same picture in WT macrophages in control condition and treated with WZ811!
Response: Sorry! This is a mistake in combining figures among different groups. We have corrected them in the revised manuscript. Thanks a lot for your carefully reviewing.
Authors must add in legend the number of mice or field used to perform statistical analysis (n= ).
Response: We have stated this information in the figure legends. Thanks for your suggestion.
Reviewer 3 Report
The logic of this manuscript is very clear, and the experimental design is rigorous. It's also a lot of work. There are a few minor issues that need to be explained.
1. How much time did the knockout mice go through from the beginning to the determination of stable inheritance? What generation of mice are the current test results based on?
2. What is the success rate of cell transfection? In this experiment, why didn't primary liver cells from mice be knocked out?
3. The results section needs to be expressed in numbers
Author Response
The logic of this manuscript is very clear, and the experimental design is rigorous. It's also a lot of work. There are a few minor issues that need to be explained.
Response: Thank you very much. The point-by-point responses for your comments are included as following.
- How much time did the knockout mice go through from the beginning to the determination of stable inheritance? What generation of mice are the current test results based on?
Response: The miR-301a knockout mice was obtained from Prof. Ma Xiaodong’s group. And Prof. Ma is the coauthor of this paper. The homozygous knockout mice have been bred for dozens of generations with stably inherited. They have been used in Ma’s group for many years, and several previous publications are performed on this line of miR-301a knockout mice. Such as: (1) Ma X, Yan F, Deng Q, Li F, Lu Z, Liu M, Wang L, Conklin DJ, McCracken J, Srivastava S, Bhatnagar A, Li Y. Modulation of tumorigenesis by the pro-inflammatory microRNA miR-301a in mouse models of lung cancer and colorectal cancer. Cell Discov. 2015 May 19;1:15005. doi: 10.1038/celldisc.2015.5. (2) Li X, Zhong M, Wang J, Wang L, Lin Z, Cao Z, Huang Z, Zhang F, Li Y, Liu M, Ma X. miR-301a promotes lung tumorigenesis by suppressing Runx3. Mol Cancer. 2019 May 23;18(1):99. doi: 10.1186/s12943-019-1024-0. (3) Zhong M, Huang Z, Wang L, Lin Z, Cao Z, Li X, Zhang F, Wang H, Li Y, Ma X. Malignant Transformation of Human Bronchial Epithelial Cells Induced by Arsenic through STAT3/miR-301a/SMAD4 Loop. Sci Rep. 2018 Sep 5;8(1):13291. doi: 10.1038/s41598-018-31516-0. (4) Li F, Wang M, Li X, Long Y, Chen K, Wang X, Zhong M, Cheng W, Tian X, Wang P, Ji M, Ma X. Inflammatory-miR-301a circuitry drives mTOR and Stat3-dependent PSC activation in chronic pancreatitis and PanIN. Mol Ther Nucleic Acids. 2022 Jan 19;27:970-982. doi: 10.1016/j.omtn.2022.01.011.
- What is the success rate of cell transfection? In this experiment, why didn't primary liver cells from mice be knocked out?
Response: Thanks for the comments. For the first question, we detected the role of miR-301a in the migration and phagocytosis of macrophages with knockout mice, only the YY1 knockdown was achieved by siRNA transfection. During the transfection, we did not detect the success rate of cell transfection but measured the YY1 knockdown efficiency after the siRNA transfection was performed based on the reported protocols and the instructions of the kit (Fig 3E, F). In order to make this issue more clearly, we revised the description in the section of “ 2.6 YY1 siRNAs transfections”. For the second question, we did not involve primary liver cells in this study, so we did not carry out kn ocked out on its.
- The results section needs to be expressed in numbers
Response: Thanks! We have numbered the results in the revise manuscript.
Round 2
